# A Deletion Upstream of SOX10 Causes Light Yellow Plumage Colour in Chicken

**DOI:** 10.3390/genes13020327

**Published:** 2022-02-10

**Authors:** Tao Zhu, Mengchao Liu, Shan Peng, Xinye Zhang, Yu Chen, Xueze Lv, Weifang Yang, Kaiyang Li, Jianwei Zhang, Huie Wang, Haiying Li, Zhonghua Ning, Liang Wang, Lujiang Qu

**Affiliations:** 1National Engineering Laboratory for Animal Breeding, Department of Animal Genetics and Breeding, College of Animal Science and Technology, China Agricultural University, Beijing 100193, China; zhutao@cau.edu.cn (T.Z.); Leaf_zxy@outlook.com (X.Z.); ningzhh@cau.edu.cn (Z.N.); 2 Beijing Municipal Bureau of Agriculture and Rural Affairs, Beijing 100005, China; 13811000113@163.com; 3 Guiyang Municipal General Station of Animal Science, Guizhou 550081, China; december_sky@163.com (S.P.); xqcpjcs@163.com (L.W.); 4 Beijing Municipal General Station of Animal Science, Beijing 100107, China; chenyu.cncn@163.com (Y.C.); lvxueze0310@163.com (X.L.); carspstp@126.com (W.Y.); likaiyanga@163.com (K.L.); zjw7432@126.com (J.Z.); 5 College of Animal Science, Tarim University, Alar 843300, China; whedky@126.com; 6 College of Animal Science, Xinjiang Agricultural University, Urumchi 830000, China; lhy-3@163.com

**Keywords:** chicken, SOX10 gene, plumage colour, pigmentation, genome wide association study

## Abstract

Chicken plumage colour is a complex trait controlled by many genes. Herein, through Rhode Island Red (RIR) and White Leghorn (WL) F1 cross populations, the segregation of plumage color was observed in females, showing white in males, and dark red (DR) and light yellow (LY) in females. The white has been found to be caused by dominant white alleles (*I*) and the DR phenotype is attributed to a sex-linked recessive silver allele (S∗S). LY is a derived feather colour phenotype and the genetic mechanism of this is unclear. In order to explore the genetic basis for LY, we randomly selected 40 DR and 39 LY chickens for paired-end sequencing. Through the use of association analysis, we found the LY phenotype is caused by a 7.6 kb non-coding deletion near the SOX10 gene. This mutation has been reported to be responsible for dark brown plumage in chicken, and subsequent diagnostic PCR tests showed that the length of the long-range non-coding deletion is 7.6 kb instead of 8.3 kb as previously reported.

## 1. Introduction

Chickens were domesticated from the ancestral red jungle fowl (Gallus gallus), which are globally distributed as a source of eggs and meat for humans [1,2,3,4]. The long domestication history has led to many variants in the chicken genome, along with considerable phenotypic alterations. Compared to their wild ancestor, the domesticated chickens present various plumage colours and excellent production traits, with numerous breeds produced to meet human needs [5,6,7]. Previous studies have shown that plumage colour is related to the content of eumelanin and melatonin [8,9]. Melanocytes in hair follicles are the source of pigment in feathers. While eumelanin appears black and melatonin is brown, both melatonin and eumelanin are the derivatives of tyrosine [10,11]. Pigment synthesis pathways are complex and involve a large number of enzymes, with many genes reported to be related to abnormal pigment synthesis [12,13,14]. Both autosomal and sex chromosomes have been found to contain genes that cause plumage colour variations. PMEL17 was identified to be the dominant white (*I*) gene and present in White Leghorn (WL) as homozygote [15]. The sex-linked Silver (S∗S) phenotype is attributed to SLC45A2 and the Rhode Island Red (RIR) has the homozygous recessive allele (S∗N) [16]. MC1R is associated with feather pigmentation in many studies [17,18,19]. Sex-linked barring and dark brown feather colours are genetically decided by CDKN2A and SOX10, respectively [20,21].

In this study, besides the dominant white, the other two plumage colours, dark red (DR) and light yellow (LY), were observed in a RIR/WL cross population in females (Figure 1). Genome wide association study (GWAS) was performed to identify causative mutations for the colour variation. The results suggested that the SOX10 gene is responsible for the feather colour variation; however, no non-synonymous mutations were identified in the SOX10 gene. Instead, the sequencing depth differences between colour variants suggested that there is a structural variant upstream of the SOX10 gene. PCR shows a 7.6 kb deletion in LY chickens that is potentially causative for the LY variation.

## 2. Materials and Methods

We generated an F1 generation for gene mapping by crossing 100 RIR males and 2000 WL females. The intercross females showed obvious plumage colour phenotype segregation, including DR and LY, while male showed white feathers. In females, contour feathers are dark red(DR) on the surface and white for the fluff, while the contour feathers of LY chickens are all light yellow for both surface and fluff (Figure 1). In order to study the plumage variation in females, we randomly selected 40 DR and 39 LY hens for paired-end sequencing.

Blood samples were collected from ulnar veins using EDTA as an anticoagulant. DNA was extracted by TIANamp Blood DNA Kit (TIANGEN BIOTECH, Beijing, China) according to the manufacturer’s instructions. Whole genome sequencing was performed on an Illumina 2500 platform with 5X average coverage. The raw data were filtered by fastp with default parameters [22]; then, the clean data were mapped to the chicken genome (GRCg6a, GCA_000002315.5) by BWA and SAMtools [23,23]. GATK3.6 and Picard (http://broadinstitute.github.io/picard/, accessed on 27 April 2016) were then used to call and filter variants [24]. SNPs were filtered using the following rules: (a) QUAL > 30.0; (b) QD > 5.0; (c) FS < 60.0; (d) MQ > 40.0; (e) MQRankSum > −12.5; and (f) ReadPosRankSum > −8.0. Additionally, if there were more than 3 SNPs clustered in a 10 bp window, all 3 SNPs were considered as false positives and removed. The variant file was converted to plink format by VCFtools [25], and SNPs with a call rate lower than 0.8 were removed. Case-control GWAS was then performed using plink [26]. The significance threshold was set at −log10 (0.05/number of variants).

The sequencing data showed that a deletion upstream of the SOX10 gene may be responsible for the LY plumage colour in hens. The primers F: TTTGCTCCCAACCCCTCATC and R: AGCCATCGGAAAAGAAGCCA were used to amplify the chromosome 1 (GRCg6a) region from 51,034,587 bp to 51,043,419 bp, which overlapped the deletion. PCR was performed in a total volume of 25 μL containing 1 μL DNA (50 ng/μL), 12.5 μL I-5${}^{\mathrm{TM}}$ 2× High-Fidelity Master Mix (MCLAB, San Francisco, CA, USA), 1 μL (2 pmol) of each primer, and 9.5 μL ultra-pure and sterile water. The PCR was cycled at 98 °C for 2 min, followed by 35 cycles of 98 °C for 10 s, 57 °C for 15 s, and 72 °C for 80 s, with a final cycle of 72 °C for 5 min.

## 3. Results

After quality control, we obtained 16,745,104 variants from the paired-end sequencing. The significance threshold was set at 8.52. GWAS results showed that a genome region on chromosome 1 from 50.93 Mb to 51.12 Mb exceeded the significant threshold (Figure 2). This region harbors 15 genes, and the SOX10 gene is the only one functionally related to feather pigmentation [20]. We therefore, assumed that SOX10 was the gene responsible for LY plumage colour variation. We then extracted variants located within the SOX10 gene to predict their effects. However, none of the variants in SOX10 were significantly related to plumage colour. Instead, our results showed that the significant variation is distributed around the genomic region 51.05 Mb to 51.09 Mb, indicating that there could be complex genome structural variation in this region.

In order to identify the causative variant, we randomly selected 10 individuals for LY and DR respectively, and merged the sequencing data into a single file. We then mapped them to chromosome 1, and calculated the sequencing coverage for each base. We found that the coverage in DR chickens at 51.03 Mb to 51.04 Mb was higher than that of LY chickens (Figure 3a). This suggested that there is a duplication in DR chickens or a deletion in LY chickens. We therefore extracted the insertion length of paired-end reads to identify the variant type, and found that the insertion length of some reads in LY chickens was longer than 7 kb while the insert length in DR chickens was under 1000 bp (Figure 3b,c).

In order to identify the breakpoint accurately, we designed primers to amplify the deletion region. The PCR results showed that all LY chickens are homozygous for the deletion while DR chickens are heterozygous for the deletion or homozygous for wild type, which suggested that LY is a recessive trait. We then sequenced the PCR product and Blasted the sequence against the chicken genome (GRCg6a). We found the genome region from 51,035,106 bp to 51,042,744 bp is missing in LY chickens, with this region showing a novel 10 bp insertion (GGTGCGGTGA). Gunnarson et al. (2011) identified an upstream 8.3 kb deletion of SOX10 which causes dark brown plumage in red jungle fowl. The deletion in our study had similar genomic position and length to the previous study, and the two deletions showed the same 10 bp insertion [20]. In order to verify the exact length of the deletion in our study, we PCR amplified the genomic region on chromosome 1 from 51,035,006 bp to 51,043,744 bp. The assembled sequence showed that the deletion is 7638 bp in length.

## 4. Discussion

In this study, we identified a 7.6 kb non-coding deletion near the SOX10 gene as the cause of the LY phenotype. Gunnarson et al. (2011) found that an upstream 8.3 kb deletion of SOX10 gene caused the DB (dark brown) phenotype in chicken, by compare the flanking sequence we confirmed that we have reported same variant [20]. This coordinate difference comes from improved genome sequencing and assembly as Gunnarson’s study used Galgal3 as the reference genome. Further analysis found that Galgal3 harbours an excessive N-gap region 30 kb upstream of the SOX10 gene, while Galgal6 showed complete sequence at this gap. Previous studies have shown that many genes are involved in the eumelanin pathway, such as MC1R, MITF, SOX10, PAX3 etc. [27]. The SOX10 gene activates the MITF gene by regulating the expression of eumelanin, and has great impact on the neural crest development. The long-range non-coding mutations of SOX10 have been reported in many species. Domyan et.al(2014) found that SOX10 deletion is related to recessive red in pigeons [28], A SOX10 upstream noncoding deletion in transgenic mice were characterized by the near complete absence of skin pigment [29], indicating that the deletion harbours a transcriptional enhancer, with this hypothesis demonstrated in a later study [30].

Many studies have shown that there are a large number of genes controlling feather colour in the chicken genome, with different mutation combinations causing different phenotypes. The dominant white allele (*I*) in WL is exclusively associated with a 9 bp insertion in exon 10 of the PMEL17 gene [19]. Gunnarsson et al.(2007) reported that the SLC45A2 gene on chromosome Z is associated with S∗S (silver), S∗N (wild type/gold), and S∗AL (sex-linked imperfect albinism) alleles in chickens [16], and cause a specific inhibition of dominant white allele (*I*). In our WL-RIR cross population, the dominant white gene, *I*, was inhibited in females; while all the male chickens show dominant white plumage. The most likely allele combinations are that WL birds harbored −/Del (SOX10), I/I (PMEL17), and Z${}^{S*S}$/W (SLC45A2) alleles; RIR birds harbored −/Del (SOX10), i/i (PMEL17), and Z${}^{S^{*}N}$/Z${}^{S*N}$(SLC45A2) alleles; the male F1 generation harbored −/−, I/i, and Z${}^{S*S}$/Z${}^{S*N}$ alleles; the LY females harbored Del/Del, I/I, and Z${}^{S*N}$/W alleles; and the DR females harbored Wt/−, I/i, and Z${}^{S*N}$/W alleles (Figure 4).

## Figures and Tables

**Figure 1 genes-13-00327-f001:**
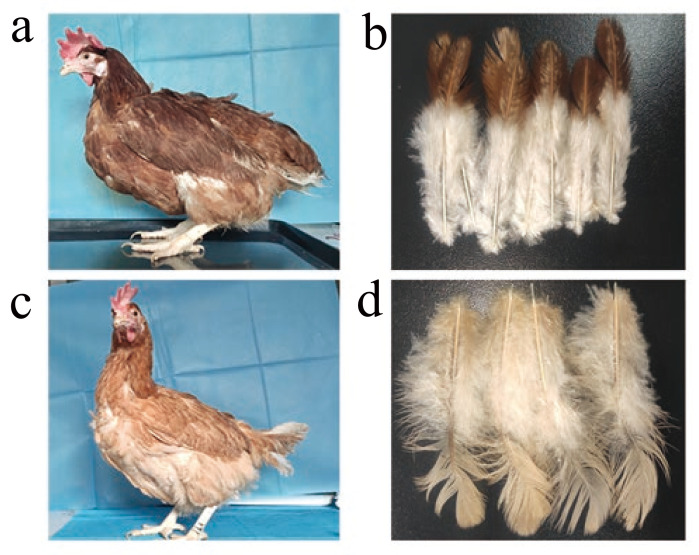
Images of chicken plumage colour patterns. (**a**) Deep red (DR) plumage. (**b**). The neck feathers of DR chicken. (**c**) Light yellow (LY) plumage. (**d**) The neck feathers of LY chicken.

**Figure 2 genes-13-00327-f002:**
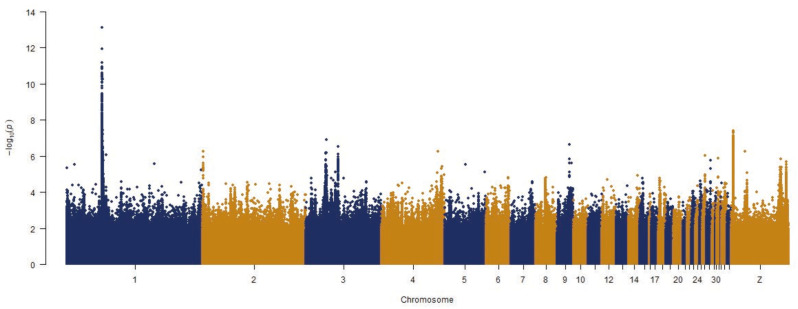
Genome-wide association analysis (GWAS) between DR and LY chickens. The horizontal axis represents the chromosome and the position, and vertical axis represents the negative logarithm of the unadjusted *p*-value for each SNP.

**Figure 3 genes-13-00327-f003:**
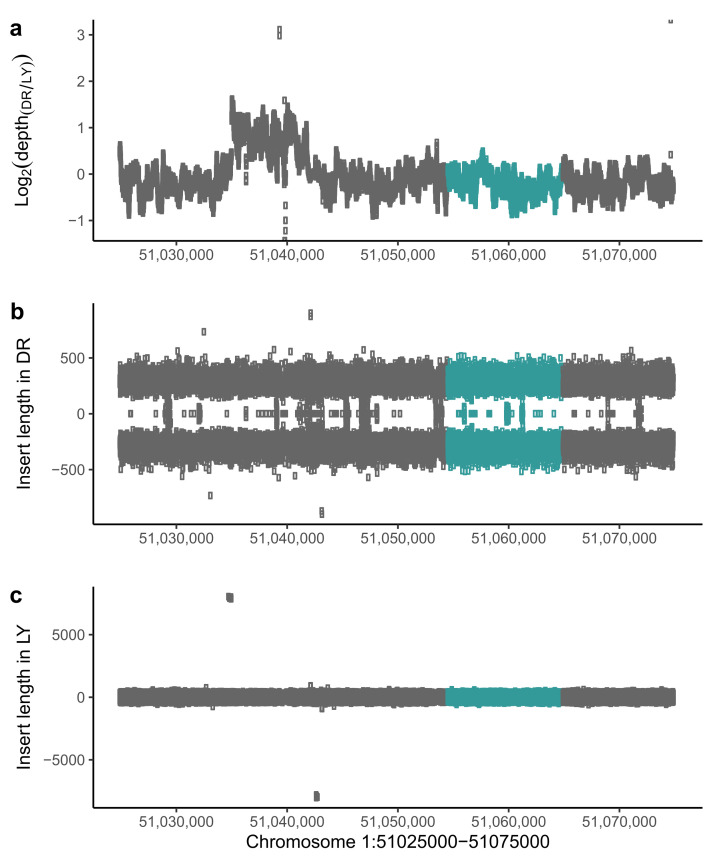
Sequencing depth and insert length showing LY chickens carrying a deletion, green dots represent the SOX10 gene. (**a**) The logarithm of sequencing depth ratio by base pair. It shows that the sequencing depth of DR chickens upstream of the SOX10 gene is higher than in LY chickens, indicating that that is a structure variant. (**b**) The insertion length of the paired reads in DR chickens, were all in the normal range. (**c**) The insertion length of the paired reads in LY chickens. The length showing part of the read insertion length is significantly higher than normal range, suggesting that the LY chickens carries a deletion upstream of the SOX10 gene.

**Figure 4 genes-13-00327-f004:**
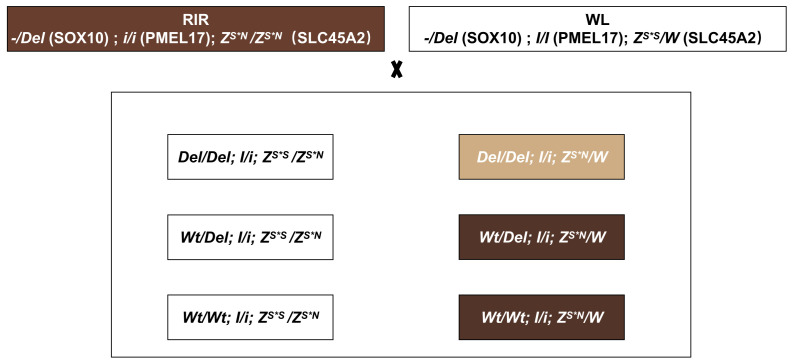
Del represents the SOX10 upstream deletion, *I* and *i* represent dominant white allelic, Z${}^{S*S}$ represent Silver (S∗S) allelic, Z${}^{S*N}$ represent wild type (S∗N) allelic. Colour in the box represent the plumage colour.

## Data Availability

The datasets generated for this study can be found in the NCBI (https://www.ncbi.nlm.nih.gov/ accessed on 27 April 2016) under BioProject ID PRJNA723465.

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
