# Peer review of "A Deletion Upstream of SOX10 Causes Light Yellow Plumage Colour in Chicken"

_genes, 2022, doi:10.3390/genes13020327_

Round 1
Reviewer 1 Report
The aim of this paper is to characterise the causative mutation(s) underlying the Light Yellow (LY) plumage phenotype in chickens. To do so, the authors conducted GWAS on 79 randomly selected Dark Red plumage chickens (n=40) and Light Yellow plumage chickens (n=39). The results show that a 7.6kb deletion and a 10bp insertion upstream of the SOX10 gene causes the Light Yellow variation in chickens.
Overall, this paper provides an important contribution to our understandings of the genetic make-up of modern domestic chicken populations by here addressing plumage colour and identifying the causative mutation underlying the Light Yellow phenotype. The methodology appears robust to me, and the results support the conclusion.
My main comment regards the discussion section. In the first paragraph, the authors highlight their findings, and mention the similarity of these findings with the work of Gunnarson et al., (2011) - also on chickens-, as well as other works on mice including those of Domyan et al., (2019). However, apart from the very last sentence of that paragraph, there is currently no interpretation as to the role the 7.6kb deletion and the 10-bp insertion they have identified may play in the Light Yellow plumage. Can any comments be made on the mechanism of action of this mutation?
Regarding the second paragraph, I don’t quite understand its relationship to the aim of the paper, or the methods and results sections as it reports on the dominant white allele (I) and the allele combinations most likely present in their selected populations. It also makes the suggestion that an allele on chromosome Z may inhibit I. This is interesting and I understand that the authors may have wished to use this finding as an opening, but given it was never mentioned previously as an aim, nor addressed in the results, this long and informative paragraph feels disjointed from the rest of the paper. If included in the paper, the authors need to better incorporate it within their aims and results.
I also have some minor comments as I find the write-up sometimes confusing, and believe the paper will benefit from some clarifications:
l.16-18: “Chickens were domesticated from red jungle fowl around 10,000 years ago.” The question of the timing of chicken domestication remains uncertain and highly debated in the literature. What the authors here present is an upper-bound date out of a variety of dates previously suggested. I believe a more accurate rendition of our current understanding of the timing surrounding chicken domestication requires a less affirmative statement and needs to highlight its uncertain nature.
l.34: Please write ‘DR’ and ‘LY’ in full as this is the reader’s first encounter with these acronyms in the main text.
The next three comments require clarifications based on whether the authors meant ‘the’ or ‘a’, these words having different implications:
l.100-101: “The deletion in our study had similar genomic position and length to the previous study, and the two deletions had a same 10-bp insertion.” Could the authors clarify if they mean both deletions included a 10-bp insertion, or if both deletions included the same 10-bp insertion. These two possibilities imply different results.
l.106-109: Could do the authors review the sentence (the structure currently doesn’t make sense), and also specify if they mean ‘reported the same variant’ or ‘reported a similar variant’. I believe it to be the latter, given it cannot be the same variant if the deletion and the 10-bp insertion is not exactly the same?
l.111: “The non-coding SOX10 upstream deletion has also been reported in mice and pigeon”. I think the authors here mean ‘A non-coding SOX10 upstream deletion […]’? If not, could the authors be more specific?
Finally, I have noted the following typos:
l.19: ‘[…] has led to many variants in the chicken genome […]’
l.33: SOX10, not Sox10
l.84: ‘In order to identify […]’
l.89-92: I think there is a word missing? Please review sentence.
l.97: “We found the genome region from 51,035,106bp to 510,427,44bp”: Typo in commas.
l.104: “[..] showed that the deletion is 7638bp in length.”
l.107-l.109: Please review sentence.
l.122: “[…], I was inhabited […]’ - please make sure ‘I’ is italicised. Same comment for l.124.
Ref 11: Formatting Error
Ref 16: There appears to be a typo in the reference
Author Response
Dear review:
Thank you for your patience, we have made modifications according to your advice, here is the detail:
1)Can any comments be made on the mechanism of action of this mutation?
The description of the mutation is displayed on the first paragraph of Discussion.
2) Better incorporate dominant white allele (I) and silver(S*S) within their aims and results.
We have streamlined this part
3) The question of the timing of chicken domestication remains uncertain and highly debated.
We changed the description to avoid this problem.
4) Word misuse
Done
Reviewer 2 Report
This paper explores the genetic basis for a plumage color polymorphism segregating in a cross between Rhode Island Red and White Leghorn. They identify a 7.6 kb deletion upstream of SOX10 and conclude that this is the same deletion as previously identified in an intercross between Red junglefowl and a strain of White Leghorn (Gunnarsson et al 2011). Thus, the paper presents a conformation of a previously reported identification of a mutation causing variation in pigmentation. The authors should make this statement already in the Abstract.
Specific comments
- I am a bit surprised that the authors used an RIRxWL F1 generation for gene mapping. The authors should report how many Dark Red and Light yellow they observed in the F1 generation.
- The results section indicated that the deletion breakpoints detected in this study and in Gunnarsson et al are indicated but there is a clear difference in the size of the deleted fragment, 8.3kb vs. 7.6 kb. The authors should analyze why there is such a difference. Is it simply that the genome assembly has changed between 2011 and 2022 so that the genome coordinates are different?
- Figure 1 legend. It is misleading to name the Dark red phenotype Wild type, just name it Dark red. It may be wild-type at the SOX10 locus but it is not a wild-type phenotype.
- Line 84 and 93, change “identity” to “identify”
- Line 95, The authors conclude that LY is a recessive trait, that implies that Dark red birds may be heterozygous for the deletion. This is in fact documented in Fig. 4 and could be stated in this sentence.
- Figure 4 is not of publication quality, but I don’t think it is necessary to include a gel image since this paper confirms a previously reported SOX10 polymorphisms. It is too much to have 5 figures in such a confirmatory paper.
Author Response
Dear review:
Thank you for your patience, we have made modifications according to your advice, here is the detail:
1) The paper presents a conformation of a previously reported identification of a mutation causing variation in pigmentation. The authors should make this statement already in the Abstract.
Done
2) The authors should report how many Dark Red and Light yellow they observed in the F1 generation.
I'm sorry we can't provide the detail as the plumage data has not been counted.
3) The authors should analyze why there is such a difference. Is it simply that the genome assembly has changed between 2011 and 2022 so that the genome coordinates are different?
We compared the flanking sequence of two assemblies, and made further explanation in Discussion part.
4) Line 84 and 93, change “identity” to “identify”
Done
5) Line 95, The authors conclude that LY is a recessive trait, that implies that Dark red birds may be heterozygous for the deletion. This is in fact documented in Fig. 4 and could be stated in this sentence.
Done
5) Figure 4 is not of publication quality, but I don’t think it is necessary to include a gel image since this paper confirms a previously reported SOX10 polymorphisms. It is too much to have 5 figures in such a confirmatory paper.
Done